# LAMBDA: Assessing Few-shot Lexical Analogical Reasoning in Language Models

## Abstract

Analogical reasoning in language models is a critical yet underexplored aspect of their capability, particularly as models grow in scale and training data. This work investigates the limitations of current models in inferring latent relational structures, focusing on lexical analogies. We introduce LAMBDA, a novel dataset of 3,000 relation-hidden lexical analogies spanning synonyms, antonyms, and derivational transformations, designed for two-shot induction. Our empirical evaluation across nine models, including four open-source models from 0.1B to 17B parameters, along with five commercial models, reveals a wide performance gap, with accuracies ranging from 0.3% to 49.3%, highlighting the challenge of systematic generalization. By analyzing error patterns such as identity echo and semantic drift, we provide insights into model weaknesses. Our findings suggest that large-scale pre-training alone does not guarantee strong relational reasoning abilities. These results identify a clear gap between lexical knowledge and reliable relation induction, and they provide a concrete target for future work on analogical abstraction in language models.

## 1 Introduction

Analogical reasoning is central to human cognition (Gentner, 1983; Hofstadter, 2001) and remains a frequently used test for vector-space semantics (Mikolov et al., 2013). Language models have grown quickly in parameter count and training data (Brown et al., 2020; Bommasani et al., 2021; Touvron et al., 2023; Meta AI, 2025), yet their ability to infer latent relational structure rather than memorize superficial patterns is disputed (McCoy et al., 2019). This question matters because tasks such as zero-shot entity linking (Logeswaran et al., 2019) and compositional question answering (Keysers et al., 2020) rely on systematic generalization.

Classic analogy benchmarks like the Google Analogy Test set (GAT; Mikolov et al., 2013) present relation labels and allow free-form decoding, letting models exploit cues or prompt tricks. Later resources such as BATS (Gladkova et al., 2016) and WordRep (Gao et al., 2014) widen relation coverage but still disclose the mapping, while ANALOGYKB supplies a million-scale resource for training (Yuan et al., 2024). Some studies suggest that larger scale does not guarantee compositional abstraction (Hupkes et al., 2020), motivating tasks that isolate a single skill.

We introduce LAMBDA (Lexical Analogy and Morphology Benchmark for Deep Abstraction), a corpus of 3,000 relation-hidden lexical analogies designed for two-shot induction. Items span synonyms, antonyms, and derivational transformations, created deterministically from WordNet (Miller, 1995) with strict length and overlap filters. This setup measures a model's ability to infer and apply an unseen mapping. It can also be viewed as a controlled in-context learning setting, because the latent relation is never named and must be inferred entirely from the two demonstrations provided in the prompt. For instance, one LAMBDA item provides two synonym pairs, `accept : admit` and `permit : allow`, and asks the model to complete `receive : ?`, whose correct answer is `take`. The same format is used for antonymy and derivational morphology, with the relation hidden in each case. Consequently, the model must infer it from the exemplars alone.

All data are derived from public sources (WordNet). No personally identifiable or sensitive content is included.

Baseline experiments reveal a steep performance ladder. Our weakest instruction-tuned baseline, Mistral-7B-Instruct-v0.3, answers only 4.9% of items correctly (45 synonyms, 70 antonyms, 32 derivations). For historical context we also tested GPT-2 medium, which achieves just 0.3%. At the high end, GPT-5.2 scores 49.3%. The observed error patterns (identity echo, surface misfire, semantic drift) mirror findings in morphological generalization and adversarial probing (Naik et al., 2018). With 3,000 trials, uncertainty is summarized using binomial standard errors and corresponding 95% Wald intervals. Recent representational studies confirm that models form internal concept vectors for relations such as antonymy yet still miss correct outputs (Opiełka et al., 2025).

LAMBDA is a lightweight CC BY-SA[1] dataset that isolates analogical abstraction without confounds from extended discourse or numeric reasoning. Initial results suggest that large-scale pre-training, although vital for lexical coverage (Liu et al., 2019), does not guarantee reliable relation induction, consistent with limits observed in compositional tests for vision-language models (Kim et al., 2023). We hope to invite exploration of richer prompting (Zhou et al., 2023), targeted fine-tuning (Lu et al., 2022), symbolic hybrids (Bogin et al., 2019), and future analysis of how models infer and apply latent relations in context.

**Contributions**

- **Benchmark:** We introduce LAMBDA, a 3,000-example dataset of relation-hidden lexical analogies balanced across synonymy, antonymy, and derivational morphology.

- **Reproducibility:** We release a deterministic generation pipeline along with JSONL data files so that any researcher can recreate the benchmark exactly.

- **Empirical evaluation:** We evaluate four open-source models (0.1B–17B parameters) and five commercial models under a strict two-shot, answer-only protocol, and report relation-wise performance on synonymy, antonymy, and derivational morphology with confidence intervals.

- **Error analysis:** We provide a clear taxonomy of failure modes (identity echo, surface misfire, semantic drift) that guides targeted model improvements.

- **Statistical rigor:** We compute binomial 95% confidence intervals for every score, enabling statistically sound comparisons with forthcoming commercial systems.

## 2 Related Work

**Analogy in cognitive science and AI.** Analogical reasoning has a long history as a proposed mechanism for human generalization and creative transfer. Classic accounts formalize analogy as structured alignment between relational descriptions rather than surface feature matching (Gentner, 1983; Gentner & Markman, 1997), with substantial experimental evidence that people can retrieve and apply cross-domain analogs when an appropriate structural correspondence is available (Gick & Holyoak, 1980). In AI, these ideas motivated explicit computational models of mapping, most prominently the Structure-Mapping Engine (SME), which operationalizes structure-mapping as an algorithmic alignment procedure (Falkenhainer et al., 1989). In parallel, a line of early AI work emphasized normative and probabilistic questions about when similarity-based transfer is warranted. Russell's analyses frame analogy as an inference procedure whose success depends on relevance and prior regularities, and provide quantitative and probabilistic treatments of similarity-driven analogical transfer (Russell, 1986a;b; 1988). Related logical approaches make the admissibility conditions for analogical inferences explicit, clarifying what must be assumed for an analogy to license a conclusion (Davies, 1987). This older work remains directly relevant to modern evaluation because it sharpens the core issue: solving an analogy requires inferring an appropriate mapping under uncertainty.

**Lexical analogy benchmarks.** Word analogy evaluation became standard with the Google Analogy Test Set (GATS) introduced alongside early embedding work (Mikolov et al., 2013). Subsequent benchmarks expanded both relation coverage and the mix of semantic and morphological categories, including BATS

---

[1] https://creativecommons.org/licenses/by-sa/4.0/

(Gladkova et al., 2016) and WordRep (Gao et al., 2014). More recent resources scale up analogy generation substantially, for example ANALOGYKB (Yuan et al., 2024), which supports training and representation learning at larger scale. These benchmarks have been widely used to compare models, but they typically either (i) make the relation effectively obvious through format and category design, or (ii) allow solutions that rely on lexical association and neighborhood structure rather than relation induction. In particular, the dominant offset-based protocol can conflate consistent relational offsets with properties of local similarity structure, which inflates apparent relational competence unless strong baselines and controls are used (Linzen, 2016). Classic formats like `man:king::woman:?` also mix multiple confounds, including frequency effects, memorized pairs, and suffix patterns, and the difficulty of many categories collapses for modern LLMs.

A line of work has therefore emphasized that headline analogy accuracy can mask redundancy, predictability, and ceiling effects, especially in slices dominated by frequent word pairs or direct word-form changes (Drozd et al., 2016; Bouraoui et al., 2020). This motivates stricter settings that reduce exploitable cues and better isolate the capability of interest. In particular, synonym and derivation categories can be inflated by ambiguity and multi-answer structure, while antonymy often degenerates into near-single-choice polarity memorization. LAMBDA targets these issues by hiding the relation label, filtering for within-item substring overlap, and requiring models to infer the mapping from two demonstrations rather than from an explicitly named category.

**In-context learning and example-based induction.** LAMBDA is closely connected to work on in-context learning, since each item requires a model to infer a latent rule from a small set of demonstrations and then apply that rule to a new query. Unlike standard few-shot prompting setups where the task may be named or described in natural language, LAMBDA withholds the relation label entirely and keeps the prompt format fixed, so performance depends on example-based induction rather than instruction following. This places the benchmark near recent work on how modern sequence models adapt from demonstrations provided only at inference time (Garg et al., 2022; Dong et al., 2024; Zhang et al., 2025). It therefore provides a controlled behavioral setting for studying a narrow form of in-context induction, hidden lexical relation inference, under minimal prompt variation. Because supervision is kept entirely in-context and task-description leakage is reduced, LAMBDA may also support future representation-level analyses of relation induction, lexical structure, and recurring failure modes such as identity echo or semantic drift. Related work has also shown that in-context learning behavior is not unique to transformer architectures, including studies of state space models and recurrent alternatives (Park et al., 2024; Nzoyem et al., 2026); we focus here on contemporary LLM baselines and leave broader cross-architecture comparison to future work.

**Analogical reasoning in language models.** Few-shot prompting has been reported to elicit analogy solving behavior in large models (Webb et al., 2022), but there is ongoing debate about whether these gains reflect systematic relational abstraction or artifacts of evaluation and prompting (Schaeffer et al., 2023; Berti et al., 2025). Complementary work probes whether relations are encoded in representations even when generation fails. Methods that constrain embeddings using lexical resources improve the geometry of synonymy and antonymy (Mrkšić et al., 2016; Sedoc et al., 2021), and probing studies show that contextual encoders can support relational queries under certain conditions (Bouraoui et al., 2020). From a computational perspective, modern LM-based analogy behavior also connects to earlier work on analogy mapping without hand-engineered symbolic representations. For example, LRME combines ideas from structure-mapping with corpus-derived relational statistics to build mappings between sets of terms, explicitly targeting relation discovery from text rather than labeled relation classes (Turney, 2008). This positions LAMBDA because it distinguishes (a) inferring a relation from limited demonstrations, (b) applying that inferred mapping to a new item, and (c) exploiting dataset regularities that reveal the intended relation.

Recent cognitive-science evaluations further this debate by directly testing robustness and transfer. Lewis and Mitchell introduce counterfactual variants of established analogy tasks designed to preserve the underlying rule while reducing similarity to likely pretraining patterns, finding sharp drops for GPT-style models while human performance remains comparatively stable (Lewis & Mitchell, 2024a). A follow-up expands the scope across multiple analogy domains and reports additional brittleness effects (for example, sensitivity to surface paraphrase and answer-order manipulations in story analogies) (Lewis & Mitchell, 2024b). Stevenson and colleagues compare LLMs to children's strategies on verbal analogies and argue that, under controls that separate relational mapping from association, LLM behavior aligns more with associative responding than

adult-like relational transfer (Stevenson et al., 2023). Closely related work tests near- and far-transfer in letter-string analogies by moving from familiar alphabets to unfamiliar symbol systems, where children and adults generalize readily but LLMs fail to carry over the learned mapping reliably (Stevenson et al., 2024). These results motivate relation-hidden protocols like LAMBDA that reduce category leakage, and they also motivate evaluating out-of-distribution re-encodings of the same relation, since robust analogy should survive surface changes.

At the generation level, prompting strategies such as chain-of-thought (Wei et al., 2022) and tool-augmented approaches can improve performance on complex reasoning tasks, but these also introduce additional scaffolding that obscures measurement of a model's inherent two-shot induction ability. LAMBDA therefore keeps the interface intentionally minimal to align the measured behavior with the older normative question emphasized in early AI treatments, namely whether a system can justify a mapping from sparse evidence under a relation-hidden protocol (Russell, 1986b; 1988).

**Positioning of LAMBDA.** Relative to prior analogy benchmarks, LAMBDA is intentionally minimal and relation-hidden, and it mixes three relation families (synonymy, antonymy, derivational morphology) under a uniform two-shot format. This design reduces category leakage and surface shortcuts common in earlier resources, and it better exposes the gap between lexical association and applying an inferred relation. It also makes the benchmark relevant to current work on in-context induction, since success depends on recovering a latent relation from demonstrations rather than following a named task description. As a result, models that are near-ceiling on legacy analogy sets can still exhibit substantial headroom on LAMBDA, particularly on derivational analogies where affix-level generalization is required and on synonymy where sense ambiguity and multi-answer candidate sets force sharper induction.

## 3 Our Approach

### 3.1 Problem definition

Let $\mathcal{W}$ be the set of lowercase English lemmas in WordNet (Miller, 1995) that are alphabetic and at least 4 characters long. For each word $w \in \mathcal{W}$, we define three WordNet-derived relation sets:

$$R_{\mathrm{syn}}(w) = \{w' \in \mathcal{W} \mid w' \text{ is a synonym of } w\},$$

$$R_{\mathrm{ant}}(w) = \{w' \in \mathcal{W} \mid w' \text{ is an antonym of } w\},$$

$$R_{\mathrm{der}}(w) = \{w' \in \mathcal{W} \mid w' \text{ is derivationally related to } w\}.$$

The task is a relation-hidden two-shot lexical analogy problem. Given two support pairs that follow the same unknown relation $r \in \{\mathrm{syn}, \mathrm{ant}, \mathrm{der}\}$,

$$(A, B), \ (C, D) \quad \text{with } B \in R_r(A), \ D \in R_r(C),$$

the model is asked to complete the query pair $(E, ?)$ by producing a single-word answer $\hat{F}$.

During dataset construction, each item contains one sampled target $F \in R_r(E)$ so that the query pair is instantiated as $(E, F)$. During evaluation, however, the model receives credit for any answer $\hat{F} \in R_r(E)$. In other words, generation samples one valid completion, while scoring is based on membership in the full valid answer set.

### 3.2 Dataset generation

Algorithm 1 constructs the dataset deterministically until each relation contributes exactly 1,000 valid instances, for a total of 3,000 analogies. Listing 1 provides the full Python generation script.

---

**Algorithm 1** Deterministic item generation

---

1: fix PRNG seed 42
2: **for** $r \in \{\mathrm{syn}, \mathrm{ant}, \mathrm{der}\}$ **do**
3:     $\mathcal{D}_r \leftarrow \emptyset$
4:     **while** $|\mathcal{D}_r| < 1{,}000$ **do**
5:         sample distinct $A, C, E \in \mathcal{W}$
6:         sample $B \sim \mathrm{uniform}(R_r(A))$
7:         sample $D \sim \mathrm{uniform}(R_r(C))$
8:         sample $F \sim \mathrm{uniform}(R_r(E))$
9:         **if** all six tokens are distinct and no token is a substring of another **then**
10:             add $\big((A,B),(C,D),(E,F),r\big)$ to $\mathcal{D}_r$
11:         **end if**
12:     **end while**
13: **end for**
14: **return** $\bigcup_r \mathcal{D}_r$

---

The substring filter removes cases with direct surface overlap that could make the task artificially easy. For example, we reject cases where one token contains another, such as `sing` and `singing`.

Corpus statistics, including token lengths, parts of speech, and candidate-set sizes, are reported in Appendix A.

### 3.3   Prompt construction

Each item is rendered as two support lines followed by the query:

$$A : B \quad C : D \quad E : ?.$$

We then append a fixed answer-only guard to the end of the user message, as reproduced verbatim in Appendix C. This guard instructs the model to return a single English word and nothing else. The same guard is used for all models and all experimental conditions.

No chain-of-thought or additional natural-language instruction is requested beyond this answer-only guard.

### 3.4   Inference protocol

Models are queried with greedy decoding, using temperature 0 and `max_new_tokens=2`, and evaluation is based on the model's normalized first word. This setup is intended to isolate lexical relation induction under a fixed answer-only protocol rather than performance under prompt variation. More specifically, the goal is to study hidden relation induction under a standardized in-context format rather than to maximize each model with model-specific variations or prompt engineering.

For GPT-5.2, we use the API's reasoning mode with `reasoning_effort=medium`, while keeping the same answer-only constraint and the same single-word scoring rule used for the other models. All other models are evaluated under the same shared answer-only protocol without additional reasoning scaffolding, so GPT-5.2's reasoning mode should be interpreted as part of the model configuration exposed by OpenAI rather than as a separate prompting method.

### 3.5   Scoring

Let $\hat{F}$ be the normalized first word produced by the model.[2]

---

[2] The normalized first word is the first alphabetic word after removing punctuation and prefixes such as "Answer:" or "The answer is." For example, "Answer: Sad" is normalized to "sad." This yields a uniform comparison across models that differ in formatting behavior.

An item is scored as correct if the predicted word belongs to the full valid answer set for the query word under relation $r$:

$$s(\hat{F}, E, r) = \mathbf{1}\big[\hat{F} \in R_r(E)\big].$$

Overall accuracy is

$$\hat{p} = \frac{1}{3,000} \sum_{i=1}^{3,000} s_i.$$

If a query word has multiple valid answers, the model receives full credit for returning any one of them. For example, if the relation is synonymy and the query word has several WordNet-valid synonyms, any member of that set is marked correct.

For summary uncertainty, we report the binomial standard error

$$\text{SE}(\hat{p}) = \sqrt{\frac{\hat{p}(1 - \hat{p})}{3,000}},$$

and the corresponding 95% Wald interval

$$\hat{p} \pm 1.96 \, \text{SE}(\hat{p}).$$

### 3.6 Output collection

We report results for four open-source and five commercial models, spanning approximately 0.1B to 17B parameters, queried through the HuggingFace Inference API and the OpenAI API. Each run logs the model output and item-level correctness for later aggregation.

Human evaluation was conducted on a randomly selected subset of 300 items, with 100 items from each relation type. The human evaluator received the same two-shot, relation-hidden format used for the model runs and was instructed to provide a single-word answer for each query. The evaluator was not told the relation type, was asked to avoid multi-word expressions unless they were natural fixed expressions, and was allowed to consult dictionary definitions but no other resources. Human responses were scored with the same normalization and WordNet set-membership rule described in Section 3.5.

## 4 Experiments and Analysis

### 4.1 Baselines

Table 1 shows the four open-source checkpoints and five commercial checkpoints we ran tests upon. Uncertainty is summarized using the binomial standard error and corresponding 95% Wald interval described in Section 3.5.

**Comparison to human performance.** To contextualize the model results, we evaluated one human participant on a randomly selected subset of 300 analogies, with 100 per relation type. The human achieved an overall accuracy of **52.7%**, with scores of 47.0% on antonyms, 66.0% on synonyms, and 45.0% on derivations. The corresponding 95% Wald interval half-widths are ±5.65 percentage points overall, ±9.28 percentage points for synonyms, ±9.78 percentage points for antonyms, and ±9.75 percentage points for derivations. Because the human evaluation uses a 300-item subset, direct model-versus-human comparison should be made on that same subset; we report model performance on the human subset in Table 2 below.

The benchmark uses a strict single-word answer protocol to measure lexical relation induction under a fixed answer-only format. This choice keeps the evaluation uniform across models and avoids conflating relation inference with longer-form generation behavior. At the same time, some WordNet-valid answers are multi-word expressions. In LAMBDA, about 10.5% of valid target sets contain a multi-word expression, including

---

[3] Parameter counts for these models are not publicly available.

Table 1: Accuracy on LAMBDA.

| Model | Params | Overall | Syn | Ant | Der |
|---|---|---|---|---|---|
| Human Evaluation | — | 52.7% | 66.0% | 47.0% | 45.0% |
| GPT-2 medium (Radford et al., 2019) | 0.3B | 0.3% | 0.3% | 0.4% | 0.2% |
| Mistral-7B-Instruct (Jiang et al., 2023) | 7B | 4.9% | 4.5% | 7.0% | 3.2% |
| Llama-4 Scout-17B (Meta AI, 2025) | 17B | 40.4% | 42.1% | 46.9% | 32.1% |
| Llama-4 Maverick-17B (Meta AI, 2025) | 17B | 46.4% | 44.8% | 48.9% | 45.6% |
| GPT-4o[3] (OpenAI, 2024) | — | 31.3% | 23.7% | 37.8% | 32.4% |
| GPT-4.1[2] (OpenAI, 2025) | — | 32.2% | 23.4% | 38.0% | 35.3% |
| GPT-4.1 nano[2] (OpenAI, 2025) | — | 22.7% | 17.3% | 30.5% | 20.3% |
| Gemini 2.5 Pro[2] (Google DeepMind, 2025) | — | 24.2% | 22.5% | 25.6% | 24.6% |
| GPT-5.2[2] (OpenAI, 2025) | — | 49.3% | 35.2% | 69.3% | 43.4% |

28.6% of synonym items. We therefore report the main results under the fixed single-word protocol and treat the filtered analysis below as a sensitivity check rather than a replacement metric.

We additionally evaluate a reasoning model, GPT-5.2, with reasoning_effort=medium under the same two-shot format, greedy decoding, and answer-only scoring. GPT-5.2 achieves 49.3% overall accuracy, with 69.3% on antonyms, 35.2% on synonyms, and 43.4% on derivations. The corresponding 95% Wald interval half-widths are $\pm 1.79$ percentage points overall, $\pm 2.96$ percentage points for synonyms, $\pm 2.86$ percentage points for antonyms, and $\pm 3.07$ percentage points for derivations. This places GPT-5.2 near the human subset score overall, though direct model-versus-human comparison should be based on the aligned subset results reported in Table 2.

**Auxiliary evaluation on the human subset.** To provide a direct comparison with the human evaluation, we also evaluated several models[4] on the same 300 items used in the human study. These auxiliary subset results are intended to complement the main 3,000-item benchmark in Table 1, rather than replace it, and to show how performance changes when the output constraint is relaxed.

On the same 300-item subset, allowing multi-token outputs changes accuracy substantially for some models. GPT-4o rises from 34.7% to 49.3%, GPT-4.1 rises from 31.7% to 56.0%, GPT-4.1 nano rises from 24.3% to 43.7%, GPT-5.2 changes from 50.3% to 46.0%, and Mistral-7B-Instruct changes from 6.7% to 8.0%.

To estimate this effect on the full benchmark, we also report an approximate filtered-set sensitivity estimate based on excluding such items. On the 300-item human subset, however, we additionally report direct multi-token evaluations in Table 2. Using our corpus-level estimate that 10.5% of target sets include a multi-word expression, the retained set has size $N = 2,685$. Rather than re-running evaluation on an explicitly filtered subset, we report an approximate rescaled accuracy obtained by dividing each model's observed number of correct items by 2,685. Equivalently, this rescales the reported overall accuracies by a factor of $3000/2685 \approx 1.117$. This quantity should be interpreted only as a sensitivity analysis for the fixed single-word protocol, not as a replacement main metric. The resulting rescaled overall accuracies are: GPT-2 medium 0.3%, Mistral-7B-Instruct 5.5%, Llama-4 Scout-17B 45.1%, Llama-4 Maverick-17B 51.8%, GPT-4o 35.0%, GPT-4.1 36.0%, GPT-4.1 nano 25.4%, Gemini 2.5 Pro 27.0%, and GPT-5.2 55.0%.

We also tested whether the single-word results are sensitive to mild stochastic decoding. Across three repeated runs at temperatures 0.1 and 0.2 on the 300-item human subset, mean overall accuracies differed only slightly from temperature 0, and standard deviations across repeats remained small.

---

[4]Gemini 2.5 Pro is not included in this auxiliary table because it was no longer available through the Gemini API at the time of these follow-up runs.

[5]GPT-5.2 was only evaluated at $T = 0$ because its API configuration did not support nonzero-temperature runs.

Table 2: Auxiliary evaluation on the 300-item human subset. Values at $T = 0.1$ and $T = 0.2$ are means over repeated runs.

| Model | Setting | Overall | Syn | Ant | Der |
|---|---|---|---|---|---|
| GPT-4o | single-word, $T = 0$ | 34.7% | 30.0% | 40.0% | 34.0% |
| GPT-4o | multi-token, $T = 0$ | 49.3% | 51.0% | 56.0% | 41.0% |
| GPT-4o | single-word, $T = 0.1$ | 33.9% | 28.3% | 37.3% | 36.0% |
| GPT-4o | single-word, $T = 0.2$ | 33.0% | 27.7% | 38.0% | 33.3% |
| GPT-4.1 | single-word, $T = 0$ | 31.7% | 22.0% | 36.0% | 37.0% |
| GPT-4.1 | multi-token, $T = 0$ | 56.0% | 54.0% | 65.0% | 49.0% |
| GPT-4.1 | single-word, $T = 0.1$ | 32.2% | 22.7% | 36.3% | 37.7% |
| GPT-4.1 | single-word, $T = 0.2$ | 33.3% | 23.3% | 37.7% | 39.0% |
| GPT-4.1 nano | single-word, $T = 0$ | 24.3% | 20.0% | 30.0% | 23.0% |
| GPT-4.1 nano | multi-token, $T = 0$ | 43.7% | 29.0% | 62.0% | 40.0% |
| GPT-4.1 nano | single-word, $T = 0.1$ | 24.3% | 19.7% | 29.7% | 23.7% |
| GPT-4.1 nano | single-word, $T = 0.2$ | 24.9% | 20.3% | 31.0% | 23.3% |
| GPT-5.2[5] | single-word, $T = 0$ | 50.3% | 34.0% | 72.0% | 45.0% |
| GPT-5.2 | multi-token, $T = 0$ | 46.0% | 35.0% | 65.0% | 38.0% |
| Mistral-7B-Instruct | single-word, $T = 0$ | 6.7% | 7.0% | 11.0% | 2.0% |
| Mistral-7B-Instruct | multi-token, $T = 0$ | 8.0% | 9.0% | 11.0% | 4.0% |
| Mistral-7B-Instruct | single-word, $T = 0.1$ | 6.6% | 7.0% | 10.7% | 2.0% |
| Mistral-7B-Instruct | single-word, $T = 0.2$ | 6.7% | 7.0% | 11.0% | 2.0% |

We also examined prompt sensitivity using Llama-4 SCOUT-17B. Using the same two-shot exemplars, a minimal format with only the analogies and a question mark, with no instruction, yielded 0.6% accuracy, while adding a natural language instruction to "infer the missing word" raised accuracy to 23.7%. Our default format with an answer-only cue, described in Section 3.3, achieved 40.4%. Thus, explicit instructions and answer constraints substantially improve output validity, but even the best prompted score remains well below human-level performance. This suggests that prompt format affects response validity more strongly than it changes underlying task performance.

## 4.2  Relation difficulty

Antonyms are the easiest relation for the smaller baselines we evaluate, possibly because polarity pairs such as *hot–cold* and *increase–decrease* are frequent and distributionally salient in pretraining corpora. Synonyms pose a similar challenge to derivations for the 17 B Maverick checkpoint, but derivations remain hardest for the 7 B Mistral model, suggesting greater morphological abstraction is needed. The noun-heavy skew in synonym and derivation queries versus the adjective tilt in antonyms (Figure A.2 in Appendix A) aligns with this pattern: polarity adjectives are short, frequent, and more easily matched, whereas derivational morphology often requires longer stems or suffix manipulations.

## 4.3  Error taxonomy

An analysis of model failures reveals three dominant patterns. Let $R_r(q) = \{w_1, w_2, \ldots, w_n\}$ denote the candidate set of target words, from which the model must select one correct word for a given query $q$.

- **Identity Echo:** The model repeats the query token $q$ instead of generating a word from $R_r(q)$. For example, given $q = cat$, the model outputs *cat*.

- **Surface Misfire:** The model applies an irrelevant form change to $q$ (e.g., pluralization), resulting in $q'$, which is not in $R_r(q)$. For example, given $q = cat$, the model outputs *cats*.

- **Semantic Drift:** The model generates $w$, which is semantically related to $q$ but not in $R_r(q)$, indicating a near-miss. For example, given $q = happy$ (expecting an antonym), the model outputs *cheerful.*

While all of the above are counted as errors in our evaluation, the third category represents outputs that are plausible in meaning (e.g., *cheerful* is related to *happy*) but do not satisfy the exact relation required. To understand the prevalence of these failure modes, we quantitatively analyzed Llama-4 SCOUT-17B's errors. We found that approximately 32% of its mistakes were semantic drifts (semantically appropriate yet not in the target set), 19% were surface misfires, and 14% were identity echoes, with the remaining 34% being uncategorized incorrect answers. This shows that a majority of model errors follow clear linguistic patterns, underscoring the interpretability of these failure modes.

### 4.4 Length ablation

Beyond aggregate accuracy, we examined whether query-word length is associated with success under the fixed relation-hidden, two-shot protocol. Because LAMBDA provides minimal contextual support, even basic lexical properties of the query token may affect whether a model can infer and apply the intended relation.

For SCOUT-17B, accuracy increases with query-word length (Spearman $\rho = 0.52$; see Figure A.1 in Appendix A for the underlying distribution). To reduce noise from sparsely populated bins, we also recomputed the correlation after excluding lengths with fewer than five examples, which increases the Spearman coefficient to 0.68. For SCOUT-17B, mild temperature changes had only a small effect on length-binned accuracy. Separate repeated-run robustness checks on the 300-item human subset are reported in Section 4.1. Figure 1 shows the length-accuracy relationship.

One possible explanation is that longer query words may contain more specific lexical or morphological cues, though this interpretation should be treated cautiously given the sparse counts in the longest bins.

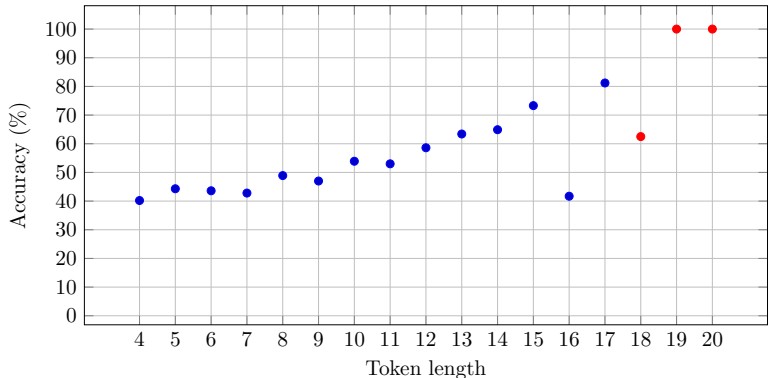

Figure 1: SCOUT-17B accuracy by query length. Spearman $\rho = 0.52$ on the full set of bins and $\rho = 0.68$ after excluding bins with fewer than five examples. The longest bins are marked in red.

These results indicate a positive association between length and accuracy for SCOUT-17B, though the trend is not strictly monotone. The bins at lengths 16, 18, 19, and 20 contain very few examples, as reported in Appendix A.1, so their accuracies should be interpreted cautiously. Accordingly, we report correlations on both the full set of bins and a filtered set excluding bins with fewer than five examples.

## 5 Discussion

### 5.1 Overview of Empirical Findings

Our results trace a variable performance ladder on LAMBDA, spanning three orders of magnitude in parameter count. GPT-2 medium (0.3 B parameters) answers only 9 of the 3,000 analogies correctly (0.3%), and even a modern 7B instruction-tuned checkpoint solves fewer than 5% of items. In contrast, the two Llama-4 17B variants both surpass the 40% threshold, indicating that large-scale pre-training and instruction tuning are prerequisites for lexical-level relational abstraction.

While the raw numbers confirm long-standing observations that analogy is brittle for small models (Drozd et al., 2016), they also coincide with evidence that seemingly "emergent" behaviours surface abruptly once models cross a certain scale. Recent work cautions, however, that such breakpoints may partly reflect metric granularity rather than genuine phase transitions (Schaeffer et al., 2023). Taken together, our baselines position LAMBDA as a sensitive analysis of the middle-to-upper portion of today's model zoo, capable of separating near-state-of-the-art open-source checkpoints that look indistinguishable on headline leaderboards.

### 5.2 Relation-wise Performance

In addition to length, we were curious whether grammatical category influences analogical performance in the same constrained setting. Part of speech provides a coarse proxy for how relational knowledge is organized in language models, since different categories encode meaning through distinct morphological and syntactic patterns. We suspected that relations involving content words with clearer inflectional or semantic structure would be easier to abstract than those involving function words or categories with weaker morphological cues, motivating a breakdown of accuracy by part of speech.

Table 3: POS (part of speech) counts for query words (Penn tags grouped).

| POS | Synonym | Antonym | Derivation | Total |
|---|---|---|---|---|
| Noun | 807 | 542 | 764 | 2 113 |
| Verb | 6 | 11 | 9 | 26 |
| Adjective | 88 | 216 | 109 | 413 |
| Adverb | 26 | 83 | 10 | 119 |
| **Total** | 1 000 | 1 000 | 1 000 | 3 000 |

As Table 3 shows, synonym and derivation queries are overwhelmingly noun-heavy, whereas antonym queries skew heavily toward adjectives.

**Antonym performance.** Across all checkpoints, antonym questions are the easiest slice (216 adjective queries). Even GPT-2, a comparatively primitive model, solves a few here, and Mistral-7B reaches the 7% mark. This mirrors psycholinguistic evidence that polarity pairs (e.g. *hot–cold*) are acquired early and occur disproportionately in text, giving models strong distributional cues. Counter-fitting work further shows that antonym relations are linearly separable in embedding space (Mrkšić et al., 2016), which may explain their accessibility to small LMs.

**Synonym performance.** Performance on synonym analogies grows more slowly with scale: Scout improves over Mistral by 38 pp, yet Maverick gains only another 2 pp. Reflecting their noun-dominated composition (807 nouns), synonyms suffer from lexical ambiguity, WordNet synonym sets often span multiple senses, forcing disambiguation from minimal context.

**Derivational performance.** Derivational morphology remains hardest overall. Although derivations also exhibit a strong noun majority (764 nouns), they include more verbs (9) and adjectives (109), introducing added morphological variation. The gap between synonyms and derivations flips sign between Scout

and Maverick, hinting that beyond roughly 10 B parameters, models begin to learn affix-level regularities (Vylomova et al., 2017). Yet Maverick still misses more than half of the derivational items, echoing recent morphology-specific evaluations that document sizeable headroom even for GPT-class systems (Romanov & Khusainova, 2023).

**Summary across relation types.** The sub-slice scores suggest a competence hierarchy:

$$\text{Antonymy} > \text{Synonymy} \approx \text{Derivation (small models)}, \quad \text{Antonymy} \approx \text{Derivation} > \text{Synonymy (large models)}.$$

Early gains reflect frequency and surface cues; later gains reflect emerging morphological abstraction. Further ablating part-of-speech (Figure A.2) and candidate-set size (Table A.3) should disentangle frequency from compositional complexity.

### 5.3 Scaling Law Extrapolation

Classic scaling laws predict log-linear improvements in accuracy with increasing parameter count (Kaplan et al., 2020). However, our results depart from this trend. Llama-4 Maverick-17B achieves the second highest score (46.4%), substantially outperforming GPT-4o and GPT-4.1 (31.3% and 32.2%, respectively), even though the latter are widely presumed to be much larger and generally stronger models.

This discontinuity suggests that model size alone is a poor predictor of relational analogy induction; data curation, architecture, or specific training regimes may matter more than scale in this setting. Our findings echo recent work questioning the universality of scaling laws and the nature of "emergent" abilities in LLMs (Schaeffer et al., 2023). Overall, scaling up does not guarantee robust lexical relational abstraction, and highly tuned open-source models may surpass commercial systems on tasks outside the typical benchmark suite.

### 5.4 Broader Impact

Systematic analogical reasoning is a central component of human language understanding, with broad implications for tasks such as scientific discovery, education, and knowledge transfer. By developing LAMBDA, a minimal-shot lexical analogy benchmark, we provide researchers and practitioners with a targeted way to assess whether language models move beyond surface pattern matching toward relational inference. While the current results reveal that even state-of-the-art models have not mastered these capabilities, a clearer diagnostic benchmark can help focus community efforts on true abstraction rather than superficial accuracy.

This work may also inform research in linguistic theory, cognitive modeling, and educational technology by clarifying which relational skills remain out of reach for current systems. At the same time, exposing specific weaknesses in analogy-making helps mitigate the risk of overestimating language model generalization in downstream applications such as question answering, knowledge graph completion, or scientific information extraction. More rigorous evaluation of relational abstraction is a step toward safer and more interpretable AI, especially as models are deployed in high-stakes or high-impact language tasks.

## Limitations and Future Directions

The present version of LAMBDA provides a targeted but partial view of analogical reasoning in language models. The design choices, English-only vocabulary, WordNet-derived relations, strict single-token scoring, and uniform two-shot prompts, were made to support interpretability, reproducibility, and statistical clarity. These same choices also define what the benchmark does and does not measure.

One limitation is coverage. LAMBDA is English-only and relies on WordNet-derived lemmas, which leaves open questions about model performance in morphologically rich languages, low-resource settings, informal registers, and colloquial or domain-specific vocabulary. As a result, models may face a narrower distribution of analogy problems here than they would on broader or more specialized terminology. Future extensions could incorporate multilingual or code-switching analogies, along with additional lexical resources, to test whether the same difficulty patterns persist beyond English WordNet.

Potential data leakage is also an important limitation. Because LAMBDA is constructed from WordNet-derived relations, it is possible that some models were exposed during pretraining to WordNet itself, derivative resources, or other lexical datasets that encode overlapping synonym, antonym, or derivational information. The benchmark therefore does not establish that models solve these items without any prior exposure to the underlying lexical relations. Rather, it measures whether a model can apply those relations in a relation-hidden two-shot format. Our deterministic generation pipeline does not remove this risk, but it does make every item fully auditable and exactly reproducible, which supports careful inspection of possible overlap and makes future filtering or deduplication straightforward.

Another limitation lies in the evaluation protocol. The fixed single-token answer format keeps scoring uniform across models, but it excludes some valid outputs, especially when a candidate set contains multi-word expressions. In the auxiliary 300-item evaluation in Section 4.1, allowing multi-token outputs improved accuracy for several models, indicating that output constraints account for part of the error under the main protocol. Additionally, the human evaluation used a single participant on a 300-item subset, so it provides initial context rather than a strong estimate of human variability. Future work could compare the current setup with freer-form evaluation and include multiple human participants with inter-annotator agreement.

Finally, while we compare models that differ in scale and training regimen, the present analysis does not isolate which aspects of pretraining data or architecture drive analogical competence. Controlled ablations, varying one factor at a time, could test whether explicit morphological pretraining, relation-focused fine-tuning, architectural changes such as character-level representations, or cross-architecture comparisons with non-transformer in-context learners improve performance. LAMBDA may also support future representation-level studies of how models encode and apply latent relations across the demonstrations and query, especially in cases where behavior diverges despite similar prompt format.

In summary, LAMBDA is a step toward more rigorous evaluation of analogical abstraction in language models, but it captures only one slice of this broader capability. Expanding the benchmark's language coverage, answer flexibility, analogy types, and model-side controls would clarify how and when language models acquire reliable analogical behavior.

## 6 Conclusion

This study introduces LAMBDA, a benchmark designed to evaluate few-shot lexical analogical reasoning in language models. We constructed a dataset of 3,000 relation-hidden analogies spanning synonyms, antonyms, and derivational morphology, and used it to systematically assess both open and closed models across a range of scales. Under the main single-word benchmark, no model surpassed 50% overall accuracy. On the aligned 300-item subset, allowing multi-token outputs substantially improved several results, indicating that answer-format constraints account for part of the gap under the main protocol. We analyzed the influence of query length and identified recurring error patterns (identity echo, surface misfire, semantic drift), confirming that major challenges persist, particularly for synonyms and derivations, and that systematic generalization remains elusive even in today's strongest models.

Although language models have advanced rapidly, there is still a substantial gap between surface pattern recognition and reliable analogical reasoning. LAMBDA enables fine-grained comparison between models that otherwise appear similar on standard benchmarks, and it motivates the development of more specialized evaluation protocols and training strategies. The auxiliary subset experiments also show that answer-format constraints matter substantially, so future work should distinguish more carefully between failures of relation induction and failures induced by output restrictions. The benchmark is also relevant to ongoing work on in-context induction, since it isolates a narrow hidden-relation setting where future behavioral and representation-level analyses can be brought into closer contact.

Overall, our results emphasize the need to go beyond aggregate performance metrics toward more rigorous evaluations of abstract reasoning. By releasing LAMBDA, we provide a benchmark for studying where current language models still fail on relation-hidden lexical analogy tasks. Our findings highlight a persistent discrepancy between human-level and model-level analogical reasoning, underscoring the value of targeted benchmarks like LAMBDA in driving progress toward more general and reliable AI language capabilities.

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

## Appendix A   Dataset Statistics

### A.1 Token-length distribution

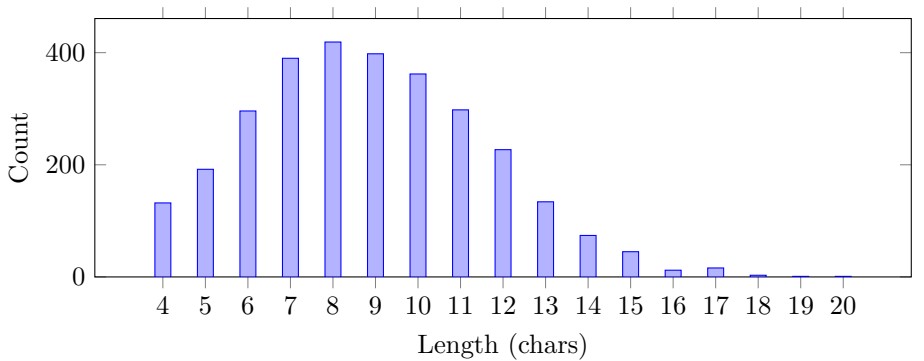

Figure A.1: Histogram of query-word lengths. Median = 9, 90th percentile = 12.

Table A.1: Counts of query-word lengths.

| Length | 4 | 5 | 6 | 7 | 8 | 9 | 10 | 11 | 12 | 13 | 14 | 15 | 16 | 17 | 18 | 19 | 20 |
|---|---|---|---|---|---|---|---|---|---|---|---|---|---|---|---|---|---|
| Count | 132 | 192 | 296 | 390 | 419 | 398 | 362 | 298 | 227 | 134 | 74 | 45 | 12 | 16 | 3 | 1 | 1 |

Table A.2: Descriptive statistics for query-word length.

| Count | Mean | Median | 90-pct | Min | Max |
|---|---|---|---|---|---|
| 3 000 | 8.85 | 9 | 12 | 4 | 20 |

Ten percent of queries contain 13–20 characters, adding morphological variety that can hinder surface memorisation.

## A.2 Part-of-speech breakdown

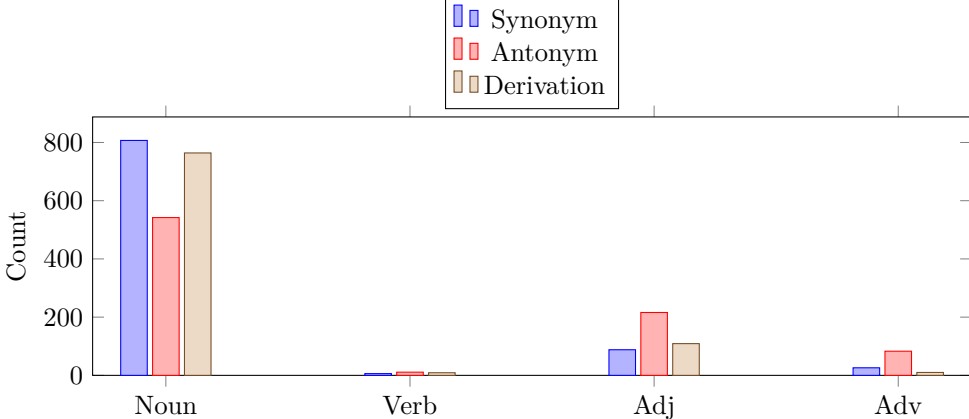

Figure A.2: POS distribution per relation.

Synonym and derivation queries are noun-heavy, while antonyms skew toward adjectives, mirroring WordNet polarity pairs such as *hot–cold*.

## A.3 Candidate-set sizes

Table A.3: WordNet candidate-set size $|R_r(E)|$ for query lemmas.

| Relation | Mean | St Dev | Min | Max |
|---|---|---|---|---|
| Synonym | 3.93 | 4.96 | 1 | 54 |
| Antonym | 1.35 | 0.71 | 1 | 5 |
| Derivation | 3.53 | 4.13 | 1 | 62 |

Antonym queries are nearly single-choice; synonym and derivation queries present much larger target sets, demanding stronger relational inference.

## Appendix B    Generation and Analysis Scripts [6]

Listing 1: Deterministic item-generation script

```
1  import json, random
2  from nltk.corpus import wordnet as wn
3
4  def get_synonyms(w):
5      s = set()
6      for syn in wn.synsets(w):
7          for l in syn.lemmas():
8              if l.name().lower() != w.lower():
9                  s.add(l.name().replace('_',' '))
10     return list(s)
11
12 def get_antonyms(w):
13     s = set()
14     for syn in wn.synsets(w):
15         for l in syn.lemmas():
```

---

[6]Full code will be released on GitHub upon publication.

```
16                for a in l.antonyms():
17                    if a.name().lower() != w.lower():
18                        s.add(a.name().replace('_',' '))
19        return list(s)
20
21  def get_derivations(w):
22      s = set()
23      for syn in wn.synsets(w):
24          for l in syn.lemmas():
25              for d in l.derivationally_related_forms():
26                  if d.name().lower() != w.lower():
27                      s.add(d.name().replace('_',' '))
28      return list(s)
29
30  def pick_related(word, func):
31      lst = func(word)
32      return None if not lst else random.choice(lst)
33
34  words = [w for w in set(wn.all_lemma_names())
35           if w.isalpha() and w.islower() and len(w) > 3]
36
37  random.seed(42)
38  dataset = []
39
40  for func, label in [(get_synonyms,"synonym"),
41                      (get_antonyms,"antonym"),
42                      (get_derivations,"derivation")]:
43      cnt = 0
44      while cnt < 1000:
45          chosen = set()
46          triples = []
47          for _ in range(3):
48              tries = 0
49              while tries < 50:
50                  c = random.choice(words)
51                  if c not in chosen:
52                      r = pick_related(c,func)
53                      if r and r not in chosen:
54                          triples.append((c,r))
55                          chosen.update((c,r))
56                          break
57                  tries += 1
58          if len(triples) != 3: continue
59          (A,B),(C,D),(E,F) = triples
60          if len({A,B,C,D,E,F}) < 6: continue
61          # ensure no word is a substring of another
62          if any(x in y or y in x for x in chosen for y in chosen if x != y):
63              continue
64          dataset.append({"few_shot":[{"input":A,"output":B},
65                                      {"input":C,"output":D}],
66                          "question":f"{E} : ?",
67                          "relation":label})
68          cnt += 1
69
70  with open("dataset/lexical_dataset.jsonl","w",encoding="utf-8") as f:
71      for entry in dataset:
72          f.write(json.dumps(entry)+"\n")
```

The following script computes per-length accuracies and correlation:

Listing 2: Length-wise accuracy and Spearman correlation

```
1  import pandas as pd
2  from scipy.stats import spearmanr
3
4  data = [
5      (4, 40.2), (5, 44.3), (6, 43.6), (7, 42.8), (8, 48.9),
6      (9, 47.0), (10, 53.9), (11, 53.0), (12, 58.6), (13, 63.4),
```

```
7      (14, 64.9), (15, 73.3), (16, 41.7), (17, 81.2), (18, 66.7),
8      (19, 0.0), (20, 100.0)
9  ]
10
11 df = pd.DataFrame(data, columns=["length", "accuracy"])
12 r, temp = spearmanr(df["length"], df["accuracy"])
13 print(f"Spearman r = {r:.2f}")
```

## Appendix C   Prompt Templates

**Default prompt template (main experiments).** Each dataset item is rendered as two support pairs followed by the query line:

```
A : B
C : D
E : ?
```

We then append the following answer-only guard (verbatim) to the user message:

```
Answer (only output the single word answer, absolutely nothing else).  IT
IS IMPERATIVE THAT YOU DO NOT RESPOND WITH ANYTHING OTHER THAN THE ONE WORD
ANSWER. DO NOT REPEAT THE QUESTION. DO NOT SAY 'THE ANSWER IS'. DO NOT SAY
'ANSWER:' OR 'ANSWER (OR ALTERNATIVE ANSWER/DESCRIPTION)', and do not show
your thought process.  It will only be a one-word answer, not multiple
words, phrases, or sentences.  DO NOT RESPOND WITH ANYTHING OTHER THAN THE
ANSWER. Additionally, do not add anything else from the definition or show
any of your thought process.  Do not include any part of the analogy you
are answering other than the answer.  If you output anything other than the
one-word answer, it will be marked as incorrect.  What is the answer?  The
answer is:
```

**Prompt-variant ablations (Figure/Table reference in main text).** To quantify prompt sensitivity, we also evaluate the same two-shot exemplars under two simplified variants:

```
Variant 1:  A : B
C : D
E : ?
?

Variant 2:  A : B
C : D
Given the above analogies, infer the missing word for E : ?.  Respond with
a single English word only.
```

