# OpenReview forum: "LAMBDA: Assessing Few-shot Lexical Analogical Reasoning in Language Models"
_TMLR — Rejected by TMLR_

### Review · Reviewer_yjA8 · 2026-02-24

**Summary Of Contributions:**

This paper introduces LAMBDA, a dataset of 3,000 relation-hidden lexical analogies made up of synonyms, antonyms, and derivational transformations, designed for two-shot induction. Using this data set the authors perform empirical evaluation across nine models, including four open-source models and five commercially available black box models. This evaluation reveals a wide performance gap, with accuracies ranging from 4.9% to 49.3%.

**Additional Comments:**

I believe some (or all) of this paper is LLM written.

**Audience:**

Yes

**Audience Explanation:**

I believe some individuals in TMLR's audience would be interested in knowing the findings of this paper as it provides a benchmark which shows a clear gap in LLM performance on two-shot induction of lexical analogies.

**Claims And Evidence:**

No

**Claims Explanation:**

They are some very odd presentation choices in this paper that makes me think some (or all) of it is LLM written.

Addressing these odd choices are needed to ensure the submission contain accurate, convincing and clear evidence.

Please see requested changes for further details.

**Requested Changes:**

- In section 3.1 The definitions of Rant and Rder seem to be missing.
- Algorithm 1 does not seem to match with the text specifically F seem to be a single element where in the rest of the paper (sec 3.5) it seems as if it is a set.
- Section 3.5 the approximation using to get plus/minus 1.8pp is also confusing. I would much rather see repeats with non-zero temperature to and averages over these to form confidence internals.
- The 4th paragraph in section 4.1 is very confusing, if certain examples can't be answer in a single token why are they included in the 3000 examples? Or conversely why is a single-token protocol used? Can you explain the difference between these two results and why people would be interested in the different numbers?
- Many of the diagrams seem unnecessary explaining simple concepts detailed in the text such as those in  Section 4.3 and Figure 1.
- Why is the length ablation section split into two sections?
- The paragraph titles in section 5.2 are very strange - could you provide what you mean by 'early' or 'tread water", why did you pick these words?
- 5.3 is very long a feels like, tangent to core question of the paper. It also introduced concepts and notation idealized Zipfian decay without defining the symbols.
- The experimental section of the of the paper feels light weight especially the fact that all experiments are performed using a temperature of zero and no repeats are performed.
- In general the text reads as very LLM written to the point it is a little distracting in sections for example:
1. "The dip around lengths 16 and 18, and the extreme bins at 19 and 20, likely reflect low sample sizes and idiosyncratic word-type effects." Surely you know the sample sizes here? They are in the appendix?
2. "For this reason, we recommend reporting correlations both on the full set and on a filtered set that removes sparsely populated length bins, as done above, to make the reported association more stable." You recommend reporting or this is what you did? This is non-standard way to write this.
- Data leakage should be discussed as the number one limitation in this paper, and this should be discussed in more detail. For example an explanation of what the following means in greater detail would be highly appreciated: "use of a deterministic generation pipeline makes the construction fully auditable and exactly reproducible which strengthens transparency and supports careful inspection of any potential leakage."

---

> ### Author Response · Authors · 2026-02-25
>
> We would like to thank the reviewer for the careful read and for calling out the places where the writing and presentation are unclear.
>
> We want to clarify the authorship and workflow behind this submission. This work, including the dataset construction decisions, experimental design, model evaluations, analysis, figures, and conclusions, was carried out by the authors. The paper is a resubmission to TMLR, and in preparing this version we used an LLM as a writing aid to shorten and rephrase portions of the text. In doing so, a few passages were inadvertently altered in ways that introduced inconsistencies and confusing phrasing. That is on us, and we apologize for overlooking these issues before resubmission.
>
> We are preparing an updated version that:
> 1. Corrects the inconsistencies you identified
> 2. Removes unnecessary diagrams
> 3. Strengthens the experimental reporting (We will run repetitions with non-zero temperature and report averages and variability).
> 4. Addresses the point your brought up regarding data leakage.
>
> We will upload the revised manuscript shortly and will make the revision substantially more clear and more internally consistent.

---

> ### Author Response · Authors · 2026-04-10
>
> **Response Part 1**
>
> Apologies for the late response! Below, we aim to address the feedback and inconsistencies pointed out.
>
> > "In section 3.1 The definitions of Rant and Rder seem to be missing."
>
> In the revision, we rewrote the problem-definition subsection and now explicitly define the synonym, antonym, and derivational relation sets.
>
> > "Algorithm 1 does not seem to match with the text specifically F seems to be a single element where in the rest of the paper (sec 3.5) it seems as if it is a set."
>
> In our revision, we clarified the distinction between dataset generation and evaluation. During generation, each item instantiates one sampled valid target, while during scoring the model receives credit for any answer in the full valid answer set. We now state that distinction explicitly in both the problem-definition and scoring subsections.
>
> > "Section 3.5 the approximation using to get plus/minus 1.8pp is also confusing. I would much rather see repeats with non-zero temperature and averages over these to form confidence intervals."
>
> We agree that the earlier presentation of uncertainty was unclear. In the revision, we replaced the fixed +/-1.8 pp phrasing with the explicit binomial standard error and corresponding 95% Wald interval formula. We also added repeated nonzero-temperature runs on the aligned 300-item subset and now report mean results over those repeats in the auxiliary evaluation.
>
> > "The 4th paragraph in section 4.1 is very confusing, if certain examples can't be answer in a single token why are they included in the 3000 examples? Or conversely why is a single-token protocol used? Can you explain the difference between these two results and why people would be interested in the different numbers?"
>
> To address this concern in our paper, we separated the main benchmark from the auxiliary subset analysis more explicitly. The paper now explains that the single-word protocol remains the main benchmark because it provides a uniform answer-only setting across all models, while the multi-token condition on the 300-item human-aligned subset is included as a sensitivity analysis to isolate how much of the error is tied to output-format restrictions.
>
> > "Many of the diagrams seem unnecessary explaining simple concepts detailed in the text such as those in Section 4.3 and Figure 1."
>
> In the revision, we removed unnecessary diagrams and simplified the presentation so that figures are kept only where they contribute directly to understanding the empirical results.
>
> > "Why is the length ablation section split into two sections?"
>
> We agree that the earlier organization was unnecessarily convoluted. In the revision, the length analysis is presented as one shortened coherent subsection, with the setup, caveats about sparse bins, and the figure discussion grouped together more consistently.
>
> > "The paragraph titles in section 5.2 are very strange, could you provide what you mean by 'early' or 'tread water', why did you pick these words?"
>
> In our revision, we renamed and rewrote these discussion subsections with more standard titles and clearer content.
>
> > "5.3 is very long and feels like, tangent to core question of the paper. It also introduced concepts and notation idealized Zipfian decay without defining the symbols."
>
> We agree that this section was overextended in the earlier version. In the revision, this part of the discussion was significantly reduced in length and refocused on the main empirical question. We additionally removed tangential side arguments that were not central to the paper.

---

> ### Author Response · Authors · 2026-04-10
>
> **Response Part 2**
>
> > "The experimental section of the of the paper feels lightweight especially the fact that all experiments are performed using a temperature of zero and no repeats are performed."
>
> We appreciate this point. In the revision, we expanded the experimental reporting by adding repeated runs at nonzero temperatures on the aligned 300-item subset and reporting averages over those runs. We also added auxiliary multi-token evaluations on that same subset, which makes the empirical section more informative while keeping the main 3,000-item benchmark unchanged.
>
> > "In general the text reads as very LLM written to the point it is a little distracting..."
>
> We appreciate the reviewer raising this concern. As noted in our earlier comment, parts of the previous submission had been shortened and rephrased with LLM assistance in a way that introduced inconsistencies and confusing phrasing. In the revision, those passages were rewritten by the authors to make the presentation more direct, precise, and internally consistent.
>
> > "'The dip around lengths 16 and 18, and the extreme bins at 19 and 20, likely reflect low sample sizes and idiosyncratic word-type effects.' Surely you know the sample sizes here? They are in the appendix?"
>
> We agree. In the revision, we rewrote the length-analysis discussion so that it refers directly to the sparse-bin counts reported in the appendix rather than speaking vaguely about them. We also made the caveat more explicit by reporting both the full-bin and filtered-bin correlations.
>
> > "Data leakage should be discussed as the number one limitation in this paper, and this should be discussed in more detail..."
>
> We agree that this needed fuller treatment. In the revision, we expanded the limitations section to discuss potential leakage more directly, including the fact that the benchmark is built from WordNet-derived relations and therefore does not establish absence of prior exposure. We also clarified that the deterministic generation pipeline supports auditability and reproducibility, which makes overlap easier to inspect in future work but does not remove the leakage risk itself.
>
> We hope our response has adequately addressed your concerns and feedback!

---

### Review · Reviewer_PQay · 2026-03-02

**Summary Of Contributions:**

The authors contribute LAMBDA, a 3000 sample benchmark of lexical analogies constructed from WordNet. It covers synonymy, antonymy, and derivational morphology. They evaluate a open and closed LLMs across sizes, and show LLM performance lagging behind their human baseline. Some analyses are presented to help the reader understand the dataset better.

Strengths:
- Clear writing
- Good motivation
- Task not too hard and not too easy, which is not trivial to find in NLP these days
- Useful diagnostics

Weaknesses:
- Human-vs-AI comparison has some flaws (see the next "Explain your answer" box below)
- Restricting the model to outputting a single token is odd, and limits the comparison between human and AI.
- Some inconsistencies in writing (see below).

**Audience:**

Yes

**Audience Explanation:**

Seems on topic for TMLR: folks interested in LLM generalization, lexical semantics, morphology, etc. would find value in a clean, relation-hidden benchmark and the accompanying analyses/discussion. Even if some design choices are debatable, LAMBDA is the kind of lightweight, controlled probe that can complement more complex reasoning benchmarks.

**Broader Impact Concerns:**

None.

**Claims And Evidence:**

Yes

**Claims Explanation:**

In 3.1 you define lemmas as 4 to 15 characters (inclusive), but appendix distribution shows lengths up to 20, and the generation snippet in the appendix only has len(w) > 3 without a max. Did you accidentally forget to remove the longer ones?

You present +/-1.8pp as if it’s “the” CI half-width; but it depends on p-hat and is maximized around 0.5.

In section 4.1: "Note that about 10.5% of LAMBDA’s target answers are multi-word expressions (28.6% of synonym answers); even if models could correctly generate all those multi-token answers, overall accuracy would increase by at most ∼10 percentage points, still leaving a substantial gap to human performance" - In the current set-up you only allow each LLM to output exactly one word, right? Is this just for simplicity's sake? I think that that, and the fact human evaluation is done one a subset of 300 while the LLMs are done on a set of 3000, makes it hard to say if there really is a gap (overall). Please consider allowing the LLM to use multi-token outputs, and also please add a table of AI results on the exact 300 samples that the human answered - this would make the comparisons fully convincing.

Also, the +/- 1.8PP for table 1 doesn't apply to the human row because it's smaller. What is the confidence interval there? It seems like human and GPT 5.2 overall score actually have overlapping confidence intervals. Also, because the human and models are evaluated on different items, even overlapping CIs understate the comparability problem.

Only a single human native speaker + dictionary allowed. That’s fine as a pilot baseline, but it’s not strong evidence for a “human gap” claim. What is the known variance of humans on this task? Is your human particularly good at it?

Finally, human row looks internally inconsistent with “100 per relation.” If each relation has exactly 100 items (as you wrote), accuracies should be whole-number percentages instead of 64.5, 42.1, and 47.2. What is happening here?

**Requested Changes:**

Report per-row CIs (Wilson or Clopper-Pearson), and separate human-vs-model uncertainty correctly. The +/-1.8pp statement is N=3000-specific. Critical for acceptance.

Allow multi-token outputs in an additional experimental condition (on the human-aligned 300-subset if cost is prohibitive). Critical for acceptance.

Add a table of AI results on the exact 300 samples that the human answered. Critical for acceptance.

The paper states 100 items per relation for the human subset, yet relation-wise percentages suggest otherwise. Also reconcile the stated 4 to 15 char bound with appendix stats showing up to length 20. Critical for acceptance.

Add an additional annotator and report inter-annotators variability. Strongly suggested but not critical.

---

> ### Author Response · Authors · 2026-04-10
>
> **Response Part 1**
>
> Thank you for your helpful and detailed feedback! Addressing your concerns will help us significantly strengthen our paper.
>
> > "In 3.1 you define lemmas as 4 to 15 characters (inclusive), but appendix distribution shows lengths up to 20, and the generation snippet in the appendix only has len(w) > 3 without a max. Did you accidentally forget to remove the longer ones?"
>
> Thank you for catching this inconsistency. The earlier version incorrectly described the lemma-length bound in the main text. In the revision, we corrected the dataset description so that it matches the actual generation procedure and appendix statistics. The task definition and dataset-generation description now align with the released generation script and the observed length distribution.
>
> > "You present +/-1.8pp as if it’s “the” CI half-width; but it depends on p-hat and is maximized around 0.5."
>
> In the revision, we removed the earlier fixed-interval phrasing and replaced it with the explicit binomial standard error and corresponding 95% Wald interval formula. We now describe uncertainty as score-dependent rather than as a single universal half-width, and we report the relevant uncertainty more carefully for both the main benchmark and the human subset.
>
> > "In section 4.1: 'Note that about 10.5% of LAMBDA’s target answers are multi-word expressions (28.6% of synonym answers); even if models could correctly generate all those multi-token answers, overall accuracy would increase by at most ~10 percentage points, still leaving a substantial gap to human performance' ... In the current set-up you only allow each LLM to output exactly one word, right? ... Please consider allowing the LLM to use multi-token outputs, and also please add a table of AI results on the exact 300 samples that the human answered, this would make the comparisons fully convincing."
>
> We agree that this comparison needed a more direct analysis. In the revision, we added an auxiliary evaluation on the exact same 300 items used in the human study and included a separate table reporting those aligned results. We also added a multi-token evaluation condition on that subset. These additions make the human-model comparison much more direct and also separate answer-format effects from relation-induction performance more clearly.
>
> > "Only a single human native speaker + dictionary allowed. That’s fine as a pilot baseline, but it’s not strong evidence for a “human gap” claim. What is the known variance of humans on this task? Is your human particularly good at it?"
>
> This is a fair concern. In our revision, we narrowed the human-comparison language and present the human result as a limited pilot baseline rather than as definitive evidence about a population-level human ceiling. We also added to the limitations and future-directions section noting that future work should include multiple human participants and report inter-annotator agreement or human variance more directly.
>
> > "Finally, human row looks internally inconsistent with '100 per relation.' If each relation has exactly 100 items (as you wrote), accuracies should be whole-number percentages instead of 64.5, 42.1, and 47.2. What is happening here?"
>
> Thank you for catching this. The earlier version contained inconsistent and incorrectly calculated relation-wise percentages for the human subset. In the revision, we corrected the human results so that they match the correct results and are internally consistent with the sampled subset.
>
> > "Report per-row CIs (Wilson or Clopper-Pearson), and separate human-vs-model uncertainty correctly. The +/-1.8pp statement is N=3000-specific. Critical for acceptance."
>
> We addressed this by revising the uncertainty discussion throughout the paper, removing the earlier one-size-fits-all presentation, and reporting uncertainty in a way that is consistent with the evaluation size being discussed. We now distinguish the full-benchmark uncertainty treatment from the smaller human-subset setting.

---

> ### Author Response · Authors · 2026-04-10
>
> **Response Part 2**
>
> > "Allow multi-token outputs in an additional experimental condition (on the human-aligned 300-subset if cost is prohibitive). Critical for acceptance."
>
> We implemented this request in the revision. The updated paper includes a multi-token auxiliary condition on the human-aligned 300-item subset, and the results are reported in a dedicated table.
>
> > "Add a table of AI results on the exact 300 samples that the human answered. Critical for acceptance."
>
> We implemented this request in the revision by adding an auxiliary table with model results on the exact same 300-item subset used for the human evaluation.
>
> > "Add an additional annotator and report inter-annotators variability. Strongly suggested but not critical."
>
> We agree that this would strengthen the human baseline. Because the current revision cycle focused first on the benchmark-definition issues, matched-subset comparisons, and auxiliary multi-token experiments, we leave multi-annotator human evaluation as future work. The revised limitations section now states this explicitly.
>
> We hope our response has adequately addressed your feedback!

---

> > ### Comment · Reviewer_PQay · 2026-04-10
> >
> > Thanks for your thorough response.

---

### Review · Reviewer_RStX · 2026-04-01

**Summary Of Contributions:**

This paper introduces a dataset called LAMBDA, containing 3,000 English-language lexical analogies (1000 synonyms, 1000 antonyms, and 1000 derivational morphology tasks). The dataset is built deterministically using WordNet and is designed for two-shot, in-context learning evaluation. The paper tests 9 models (ranging from GPT-2 to frontier models like GPT-5.2 and Llama-4 variants) and finds that even the best models max out at around 49.3% accuracy. The authors categorize the errors into identity echo, surface misfire, and semantic drift, concluding that scale alone doesn't guarantee relational reasoning.

**Strengths:**
1) The dataset generation is deterministic, lightweight, and completely reproducible (the inclusion of the Python script is a positive thing).
2) Categorizing the types of errors (semantic drift vs. identity echo vs. surface misfire) is a nice touch that adds interpretability to the failures, rather than just reporting a flat accuracy score.
3) Rigourous default scoring function, with a large sample size of 3000 used for Wald confidence interval calculation.
4) Great lit review going in depth on congnitice science, lexical analogies, etc. That said, some limitations must be pointed out (see weaknesses).
5) The authors provide a comprehensise limitation section, which, although touches on several of my concerns below, remains ciritical on many ways.

**Weaknesses:**
1) The general methodology has glaring contradictions, specifically regarding chain-of-thoughts, output token limits and multi-word ground truths (related details on weakness number 5).
2) The human baseline score (51.3%) is very poor. This is a potential red flag indicating the dataset is fundamentally noisy, overly ambiguous, or too heavily indexed on WordNet's idiosyncrasies.
3) Furthermore and with more specificity, I wonder how much of the human's failures are due to Semantic Drift as described in page 7. If so, then the problem is perhaps with the WordNet limitation then? In general, it would be useful to probe, beyong Llama-4, how much of each type of failure model are observed for all the baselines considered. Similar performance across all baselines could be indicative of a property of the dataset, which would be useful for any reseracher/practitionner adopting LAMBDA down the line.
4) The literature review is great, but does not include in-context learning (ICL) [1], ignoring the actual mechanistic literature on how modern transformers perform it. ICL is only touched upon through the lens of few-shot prompting; which is still not enough. Furthermore, the baselines tested are Transformer-based models, although other architectures have been shown to perform ICL [3]. It would be good to see how SSMs, Weight-Space RNNs [4], etc. perform on this task.
5) Related to the previous weakness. The authors mention they did not use chain-of-thought reasoning, which is concerning, because this is a distinctive feature of modern LLMs that allows them to perform better of complex tasks such as relational induction. This omission sheds doubts into the competitiveness of the baselines (compared to the human evaluator). This said, at the same time, the authors mention that they use "reasoning" on GPT-5.2. This begs the question: what type of reasoning approach is it? Is a similar reasoning approach available for other baselines here?
6) "Small models find antonyms easiest, likely because polarity cues (e.g. hot–cold, increase–decrease) are memorized during pre-training." Why would this polarity feature be specific to **small** models ?


**References:**
- [1] Zhang et al., Training Dynamics of In-Context Learning in Linear Attention, ICML 2025.
- [2] Park et al., Can Mamba Learn How to Learn? A Comparative Study on In-Context Learning Tasks, arXiv 2024.
- [3] Nzoyem et al., Weight-Space Linear Recurrent Neural Networks, ICLR 2026.

**Audience:**

Yes

**Audience Explanation:**

Despite the methodological flaws, the community is always interested in probing the limitations of LLMs. The fact that Llama-4 Maverick-17B outperforms much larger commercial models like GPT-4o on a purely lexical task is an interesting data point that contradicts traditional scaling laws. Also, highlighting that derivational morphology is still a massive blind spot for instruction-tuned models is a valuable observation for researchers working on tokenization and morphological robustness.

**Broader Impact Concerns:**

No major concerns here.

**Claims And Evidence:**

No

**Claims Explanation:**

The central claim that models lack systematic analogical generalization is confounded by how the experiment was actually run. The evidence isn't convincing because the methodology sets the models up to fail in ways that I beleive have nothing to do with their reasoning abilities.

First, the single-token vs. multi-word contradiction is a critical flaw. In Section 3.4, you explicitly state you use "greedy decoding (temperature 0, max_new_tokens=2), forcing a single-token answer". But then you admit in Section 4.1 that 10.5% of the target answers are multi-word expressions! We cannot mathematically expect a model to get this correct. While you try to address this with a "filtered-set accuracy" later (rescaling by 3000/2685 $\approx$ 1.117), it still sounds like just a post-hoc math trick. In summary, the models were evaluated using a prompt that inherently inhibits their abilities.

This is also to say that: only the 2685 valid single-token examples could have been kept in the dataset. Is there a reason why all 3000 examples had to be left in there, depsite the fact that this creates an impossible evaluation scenario?

(I appreciate that this was addressed as a limitation.) Second, the reliance on WordNet as the absolute ground truth for evaluation is notoriously brittle. You classify "happy -> cheerful" as a "semantic drift" error when the target relation is an antonym. But what if the model outputs a perfectly valid analogy that just happens to not be in WordNet's specific candidate set? If a model generates a valid but non-WordNet synonym, your rigid scoring script marks it wrong.

This reliance on WordNet is all the more concerning because a native English speaker _with unlimited time and access to a dictionary_ only scored 51.3%. If a human baseline on a basic lexical analogy task is barely better than a coin flip, then I'm inclined to believe that the benchmark has issues. It isn't measuring a model's ability to reason; it is measuring its ability to guess which specific synonym the WordNet script randomly selected.

**Requested Changes:**

**Critical to secure acceptance:**

1. The authors should either filter the entire 3,000-item dataset to only include single-word answers before running the models, or they need to relax their decoding constraints to allow multi-token generation.
2. Investigate the Human Baseline a little deeper. I do not believe it is acceptable to just have a 51.3% human accuracy and move on. If this is due to the specific native English speaker chosen, then perhaps other humans should be considered? If this is due to WordNet's rigidity, then a fundamental reevaluation of the dataset would be required.
3. Update the literature review to include in-context learning.


**Would simply strengthen the work:**
1. Investigate the types of failure modes for each baseline.
2. Clarify weakness number 6 above.

---

> ### Author Response · Authors · 2026-04-10
>
> Thanks for your helpful feedback! Addressing your suggestions will help us significantly strengthen the quality of our paper.
>
> > "Please either filter the dataset to single-word answers or relax decoding to allow multi-token generation."
>
> We agree that this was an important issue. In the revision, we added an auxiliary evaluation of model performance on the exact 300-item human subset with multi-token outputs allowed, while retaining the original single-word benchmark as the main protocol. This lets us separate the benchmark’s original constrained setting from a more permissive follow-up condition, and it makes the role of answer-format constraints much clearer. We also revised the surrounding discussion to present the filtered-set rescaling as a sensitivity analysis rather than as a replacement main result.
>
> > "The human baseline is too low and needs deeper investigation."
>
> In our revision, we expanded the human-evaluation discussion to report the subset size, relation-wise breakdown, and separate uncertainty for the human results, and we added direct model results on the exact same 300 items. We also clarified the limits of the single-participant human baseline and noted that future work could include multiple human participants and inter-annotator agreement.
>
> > "The literature review should include in-context learning."
>
> We revised our Related Work section to include broader discussion of few-shot prompting, relational induction, robustness, transfer, and recent work on analogy behavior in language models. This was added to position LAMBDA more clearly relative to the modern in-context learning literature rather than only to older lexical analogy benchmarks.
>
> > "It would strengthen the work to investigate failure modes for each baseline."
>
> In our revision, we retained and clarified the error taxonomy, and we expanded the discussion of failure modes so that the reader can better interpret what kinds of wrong answers models produce.
>
> > "Please clarify the statement about small models finding antonyms easiest."
>
> We agree that the earlier phrasing was too broad. In the revision, we rewrote this discussion to state the pattern more precisely and to connect it to the frequency and salience of polarity pairs in pretraining corpora, rather than suggesting that antonymy is inherently easier in a general sense.
>
> We hope our response has addressed your comments!

---

> > ### Comment · Reviewer_RStX · 2026-04-17
> >
> > Thank you for updating your manuscript. I have read and I am satisfied with your changes.

---

### Decision · Action_Editor_s3n4 · 2026-06-07

**Recommendation:** Reject

**Additional Comments:**

There has been quite some discussion among reviewers, concerning the strength of the benchmark and baselines as well as on the usage of LLMs to generate the content of the paper. Ultimately, reviewers recommended  accept (2x) and weakly accept. However, a closer look prompted by the EiC revealed that LLMs were indeed used up to the point they generated hallucinated references, among which:

Berti et al., 2025 – "A survey on emergent abilities in large language models" Listed authors: Andrea Berti, Thomas Scalf, and Hongjing Lu Actual authors: Leonardo Berti, Flavio Giorgi, Gjergji Kasneci

Opiełka et al., 2025 – "Conceptvectors: A new approach to interpretability in language models" Listed: Gustaw Opiełka, Anna Rogers, Djamé Seddah, and Aleksandr Drozd Actual: Gustaw Opiełka, Hannes Rosenbusch, Claire E. Stevenson Actual name of paper: "Analogical Reasoning Inside Large Language Models: Concept Vectors and the Limits of Abstraction"

Nzoyem et al., 2026 – "Weightspace linear recurrent neural networks" – ICLR 2026 Listed: Roussel Desmond Nzoyem, Nawid Keshtmand, Kazuki Irie, Jakob Foerster, Shimon Whiteson, Razvan Pascanu, Meire Fortunato Actual ICLR 2026 version: Nzoyem, Keshtmand, Crespo-Fernandez, Tsayem, Santos-Rodriguez, Barton, Deakin

Kim et al., 2023 – "VIOLIN: Virtual observatory for language model inference" – CVPR 2023: No such paper found.

This is sign of scientific misconduct and the paper is therefore rejected.

**Audience:**

Yes

**Audience Explanation:**

The topic is clearly of interest for ML and AI researchers working on reasoning and LLMs.

**Claims And Evidence:**

No

**Claims Explanation:**

This paper introduces a novel dataset, LAMBDA, consisting of thousands of relation-hidden lexical analogies including synonyms, antonyms, etc, with the aim of assessing the abilities of large language models to deal with analogical reasoning. The major claim in the paper is that current LLMs still struggle with this kind of reasoning. The reviewers praised the introduction of the dataset, but were skeptical w.r.t. baselines, such as the human baseline which scores quite low, indicating that the dataset construction can be per se noisy or problematic.

A further in-depth look, with the help of EiC revealed that LLMs were used in a non controlled way to write the paper, and resulted in hallucinated references. See comments below. This is sign of scientific misconduct and the paper is therefore rejected.